# An analysis of adolescent leisure activity structure based on subjective well-being: Focusing on social network analysis

**Jinseok Oh**[1], **Sungmin Son**[2], **Jin-Hyuck Park**[1]*

1 Soonchunhyang Exceptional Children Institute, University of Soonchunhyang, Asan-si, Chungcheongnam-do, Republic of Korea, 2 Department of Psychiatry, Wonju College of Medicine, Yonsei University, Wonju-si, Kangwon-do, Republic of Korea

* roophy@naver.com

## Abstract

This study examines how adolescent leisure activity networks relate to subjective well-being (SWB) using Statistics Korea's 2019 Time Use Survey. The analysis includes 241 high-SWB and 241 low-SWB adolescents, assessing network density, inclusiveness, average distance, isolated nodes, degree centrality, and cohesion through NetMiner 4.0, with descriptive statistics processed in SPSS ver. 25.0. The results show clear differences in leisure activity networks. High-SWB adolescents engaged in more social activities and sports, while low-SWB adolescents participated in fewer, more solitary activities. High-SWB networks were diverse and well-connected, whereas low-SWB networks were more fragmented. Screen-based activities also played different roles: supporting social connections in high-SWB adolescents but reinforcing isolation in low-SWB adolescents. This study visually highlights that leisure participation varies by SWB level. The findings suggest that promoting diverse and interactive leisure activities can improve adolescent well-being, offering insights for policy and intervention programs.

## Introduction

Subjective well-being (SWB) refers to an individuals characteristic emotional state, encompassing satisfaction, happiness, and a sense of general well-being in daily life, making it a crucial indicator of psychological health [1]. Research has consistently demonstrated a close association between mental health and subjective well-being, revealing that challenges to SWB may pose significant risks to mental health [2]. For adolescents, whose emotional and psychological foundations are still developing, disruptions in SWB can lead to profound impacts, increasing susceptibility to issues such as depression, anxiety, and social withdrawal. Thus, interventions that support SWB during adolescence are essential for safeguarding mental health during this critical developmental stage.

**Data availability statement:** The data used in this study are publicly available from Statistics Korea (KOSTAT - https://mdis.kostat.go.kr/index.do). Specifically, the 2019 Time Use Survey dataset can be accessed through the KOSIS (Korean Statistical Information Service). The data are open to the public and can be freely downloaded for research purposes.

**Funding:** This research was supported the Soonchunhyang University Research fund and the Ministry of Education of the Republic of Korea and the National Research Foundation of Korea (NRF-2023S1A5C2A03099545). The funders had no role in study design, data collection and analysis, decision to publish, or preparation of the manuscript.

**Competing interests:** The authors declare no conflict of interest.

With growing concerns over adolescent health, there is a rising demand for social and policy attention, particularly in the area of mental health alongside physical health [3]. Adolescents today face unique and evolving challenges—shifting family dynamics, academic pressures, school violence, and exposure to potentially harmful environments—which have been associated with increasing rates of stress, anxiety, and depression [4]. According to the World Health Organization (WHO), approximately 14% of adolescents aged 10–19 worldwide had mental health issues in 2019, even before the COVID-19 pandemic exacerbated these concerns [5]. Furthermore, studies indicate that around 75% of mental health disorders that manifest in adolescence persist into adulthood [6], underscoring the importance of addressing adolescent mental health early.

Leisure activities have been associated with improved mental and physical health, along with greater life satisfaction among adolescents [7]. In recent years, leisure participation has gained attention as a means of supporting adolescent mental health [8]. Korea's survey data shows an increase in leisure activities undertaken by adolescents for health purposes, with participation rates rising from 1.2% in 2016 and 2019 to 3.1% in 2022 for youth aged 15–19 [8]. These activities are developmentally significant, enhancing quality of life, academic achievement, psychological well-being, and identity formation [4]. From a mental health perspective, leisure also plays a vital role in stress reduction and management [3].

Previous research on adolescent leisure activities has explored the effects of activity type and frequency on SWB and happiness. Latent profile analysis has identified statistically significant differences in SWB based on leisure activity patterns, suggesting that structured leisure engagement can bolster subjective well-being [9]. Studies have also documented average participation rates in leisure activities, with popular activities including spending time with friends, playing video games, and engaging in media consumption at home [9,10]. A meta-analysis examining the impact of leisure participation on SWB, utilizing 37 effect sizes and data from 11,834 individuals, found a moderate positive association between leisure participation and SWB. The study indicated that this relationship was mediated by leisure satisfaction, with leisure interventions significantly enhancing SWB levels (d = 1.02) [11].

These findings suggest that leisure activities may be crucial in fostering SWB and, by extension, supporting mental health. Additional studies, such as one examining SWB and leisure participation among Croatian citizens, have highlighted that leisure activities contribute to SWB, though patterns of activity vary across age and gender [12].

Despite these insights, prior research has predominantly relied on cross-sectional designs and quantitative analyses that measure frequency or specific metrics, limiting the ability to fully understand the intricate relationships between activities and their contributions to adolescent well-being [12].

Social Network Analysis(SNA) provides an innovative approach to addressing these research challenges by visually representing and interpreting the connections among various elements. Previous studies have primarily focused on analyzing the frequency of individual activities or specific metrics, but such approaches alone are

insufficient to fully capture the complex relationships and interactions among activities [13]. In contrast, SNA goes beyond the analysis of isolated activities and instead explores the structural relationships between activities, identifying key activities within the network. SNA allows for the examination of not only relationships between individuals but also between individuals and objects, thereby uncovering patterns and clusters within complex datasets [13].

This study employs SNA to visualize and analyze the relationships between adolescents' leisure activities and their connection to subjective well-being (SWB). Notably, unlike traditional methods that assess the impact of individual activities on well-being in isolation, SNA considers the interactions among activities and their structural characteristics within a network, making it the most suitable analytical tool for this study. In the constructed network, nodes represent individual leisure activities, while links denote associations between activities based on co-occurrence or frequency of participation [14]. This approach facilitates the identification of core activities that are strongly associated with higher levels of SWB, as well as peripheral activities that may have unique or lesser impacts [14].

Ultimately, the application of SNA provides a more comprehensive understanding of adolescents' leisure activities by demonstrating that they do not function in isolation but rather within interconnected structural relationships. By systematically analyzing the frequency and interconnections among activities, SNA offers valuable insights into the structure and characteristics of leisure activities that contribute to adolescent well-being [13,14].

The purpose of this study is to systematically investigate the relationship between adolescents' leisure activities and subjective well-being (SWB) through social network analysis (SNA). By identifying key leisure activities that contribute to the enhancement of SWB, this study aims to propose effective leisure activity strategies that facilitate balanced engagement and promote social connectivity among adolescents. Furthermore, the findings of this research will contribute to the development of evidence-based policies and intervention strategies that align with the developmental needs of adolescents, ultimately fostering mental health and resilience. Given the various challenges faced by adolescents, understanding the structural characteristics of leisure activity networks and their interaction with SWB is essential for enhancing social integration and mitigating the negative effects of social isolation and passive leisure behaviors through targeted policies and intervention programs.

The research question and hypotheses in the present study are as follows.

Research Question: What differences are observed in the structural characteristics of leisure activity networks, weekday and weekend activity patterns, and the role of screen-based activities, according to adolescents' levels of subjective well-being?

Hypotheses: First, adolescents with higher levels of subjective well-being are expected to form cohesive leisure activity networks characterized by high network density and inclusiveness and a short average path length through diverse social interactions. Second, while adolescents with higher levels of subjective well-being will utilize screen-based activities as a tool to enhance social connectedness, those with lower levels of well-being are predicted to rely on these activities, thereby reinforcing isolated behavioral patterns. Third, in terms of leisure activity networks on weekdays and weekends, adolescents with higher levels of subjective well-being are anticipated to maintain diversity and balance in their activities, whereas those with lower levels of well-being are expected to exhibit similar and restricted activity patterns.

## Materials and methods

### Subjects

This study analyzed the cohesive structures of adolescent leisure activity networks based on subjective well-being levels. To compare adolescents with high and low SWB, data were obtained from the 2019 Time Use Survey, The primary purpose of the Time Use Survey is to systematically measure and analyze how citizens allocate their time in daily life. This survey records the time spent on various activities, including leisure activities, household labor, child-rearing, and social engagement, thereby collecting statistical data on citizens lifestyle patterns and time utilization [13]. The significance of this survey lies in its utility for policy formulation, understanding social changes, economic analysis, health and well-being

research, and serving as foundational data for social statistics. These elements constitute the basis of the data, and the findings of the survey have established themselves as reliable foundational data that can be utilized across various fields [13].

The survey consists of 12 household-related items, 14 individual-related items, and 11 time diary-related items. For this study, data on adolescents aged 10–19 were extracted from the raw dataset of the 2019 Time Use Survey. The selection of variables was as follows: First, from the time diary-related items, which recorded activities performed over a 24-hour period, 49 leisure activity items, one item assessing subjective well-being, and one item assessing health status were selected from a total of 153 primary activities categorized into subcategories. Second, from the individual-related items, one item assessing leisure satisfaction was chosen. Third, from the household-related items, one item on gender, one item on date of birth, and one item on educational attainment were included. These variables were finalized for analysis in this study [13].

Referring to the study by [15], adolescents were categorized based on their level of subjective well-being using the following criteria. Adolescents with a high level of subjective well-being were defined as those who scored between 1 and 4 on the subjective well-being scale and between 1 and 2 on the health status scale. Conversely, adolescents with a low level of subjective well-being were defined as those who scored between 5 and 7 on the subjective well-being scale and between 3 and 5 on the health status scale. For both scales, higher scores indicate poorer conditions.

Leisure satisfaction is defined as the extent to which an individuals various needs are fulfilled through leisure activities and was measured using a 5-point Likert scale [16]. Similarly, health status is defined as an individuals perception and assessment of their physical, mental, and social functioning and well-being related to health and was also measured using a 5-point Likert scale [17].

This classification resulted in a final sample of 482 adolescents, with 241 classified in the high subjective well-being group and 241 in the low subjective well-being group. The selection and exclusion criteria for the study participants were as follows:

The study subjects inclusion criteria was as follows: 1) Adolescents aged 10–19, 2) Individuals meeting the criteria for mood and health status evaluation based on subjective well-being levels. The exclusion criteria was as follows: 1) Individuals who did not respond to the survey, 2) Cases of duplicate responses or incomplete responses, 3) Individuals unable to participate in leisure activities due to other factors unrelated to the research purpose (e.g., mental or physical disabilities).

The data utilized in this study is a publicly available dataset provided directly by Statistics Korea, collected in strict compliance with the Personal Information Protection Act and the data protection guidelines of Statistics Korea. This dataset originates from the Time Use Survey, which encompasses the entire Korean population and was collected with explicit participant consent during the original survey process. As the dataset was obtained as secondary data, no additional ethical approval was required for this study [13].

This survey is characterized by its high reliability and representativeness, owing to its rigorous sampling design and systematic data collection methods. This survey reflects the typical patterns of adolescent leisure activities prior to the pandemic, thereby providing a baseline dataset free from the distortions of abnormal environmental changes [13]. Moreover, the Time Use Survey emphasizes the capture of long-term social trends rather than short-term fluctuations, making it particularly suitable for understanding the fundamental structure of youth leisure behavior. Its compatibility with comparisons to more recent data and the potential for enhancement through follow-up studies further support the ongoing expansion of research. Given that previous studies have successfully utilized time use survey data from specific periods to analyze long-term social patterns [18], the academic validity of employing the 2019 dataset is well substantiated.

Furthermore, all personally identifiable information (e.g., names, addresses, resident registration numbers) was completely removed during the data collection process, ensuring full anonymization. The researchers adhered strictly to Statistics Korea's data usage guidelines in downloading and utilizing the dataset, maintaining compliance with research ethics and principles of personal data protection. Therefore, this study ensures that the data collection and utilization processes meet legal and ethical standards, affirming the commitment to data protection and research integrity [13].

## Study procedure

The study involved constructing and analyzing leisure activity networks of adolescents based on subjective well-being levels using the time use survey data, as shown in Fig 1. In this study, we utilized the 2019 dataset to perform random sampling via the Oracle database, selecting 241 participants for both the high-SWB and low-SWB groups. Following the collection of survey data, we conducted data preprocessing to structure the information in a standardized format for analyzing adolescents' leisure activity networks. Leisure activity items were manually input into Excel, and network modeling was executed using NetMiner to transform the original 2-mode network into a 1-mode network.

The 2-mode network comprised two types of nodes: individuals and leisure activities, with individual nodes connected to the activities they engaged in. To convert this into a 1-mode network, we measured the similarity between individuals participating in the same leisure activities, forming direct connections based on these similarities [19,20]. Cosine similarity was employed to calculate the degree of similarity between individuals based on their person-activity relationships. Links between individuals were established only when the cosine similarity exceeded a specific threshold, and weights were assigned to these links based on the frequency of shared activities or the time spent on them [21].

During the network transformation, criteria were set to enhance analytical reliability by removing redundant information and unnecessary nodes. Activities with only one participant were excluded, weak links with negligible cosine similarity were filtered out, and inactive nodes with minimal connections or low centrality within the network were eliminated to clarify the focus on inter-individual network structures. The resulting 1-mode network was designed to analyze how types and patterns of leisure activity participation influence the formation of relationships and structural characteristics among individuals. This analysis facilitated a systematic exploration of the relationship between leisure activity participation patterns and network relational structures.

Additionally, the cosine similarity-based network transformation method applied in this study offers a more precise measurement of the associations among adolescents' leisure activities, objectively reflecting inter-individual similarities compared to traditional simple linkage methods [22,23] This approach provides a robust foundation for in-depth analysis of the structural characteristics of adolescent leisure activity networks and a more accurate investigation into their relationship with subjective well-being. Upon completing the modeling process, we conducted social network analysis focusing on network distribution characteristics, centrality measures, and cohesion structure analysis (Table 1).

The results were visualized, and descriptive statistics were calculated to analyze general characteristics. Finally, based on [21] study, we selected network characteristic distribution indicators—namely, density, inclusiveness, mean distance, and isolated nodes—as shown in Table 1.

## Data analysis

The collected data were systematically organized and coded, and analyses were conducted using NetMiner 4.0 software. The general characteristics of the research participants were analyzed through descriptive statistics, while the structural

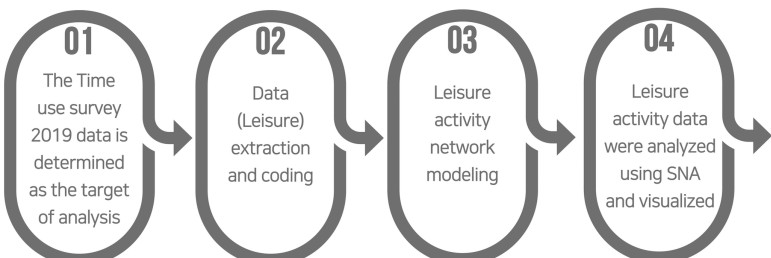

**Fig 1. Study process.**

**Table 1. Classification of network analysis index.**

| Classification | Analysis Method | Index | | Content |
|---|---|---|---|---|
| Network Characteristics Analysis | Network Level | Density | | The proportion of actual links between nodes to the total possible links in a network |
| | | Inclusiveness | | The proportion of the number of nodes that have established connections, excluding isolated nodes. |
| | Node Level | Adjacency Index | Mean Distance | The measured value of the average shortest distance between any two nodes. |
| | | Connectivity Index | Isolated Nodes | Independent nodes that have no connections with other nodes. |
| Central Structure Analysis | Network/Node Level | Centrality Index | Degree Centrality | The ratio of the total number of nodes to the number of nodes that have actual relationships. When centrality of connection is high, it indicates key nodes that are co-located with other nodes. |
| Cohesion Role Analysis | Group Level | Cohesive Structure | | By utilizing the centrality of the connections in the data, one can identify the characteristics and differences of the sub-networks. |

properties of adolescent leisure activity networks were examined using social network analysis (SNA). A significance level of $p < 0.05$ was applied to all statistical analyses.

To assess the statistical significance of the network data, Quadratic Assignment Procedure (QAP) correlation analysis was performed. Given that network data exhibit interdependencies between nodes, traditional correlation analyses are limited in their applicability. QAP correlation analysis addresses this issue by testing the significance of relationships between matrices [24,25]. This method computes correlation coefficients between two network matrices and then randomly permutes the matrices multiple times to evaluate the statistical significance of the observed relationships. Through this approach, the analysis ensures that identified patterns in the network are not the result of random occurrences but rather represent meaningful structural relationships [26].

The application of QAP correlation analysis in this study was based on the following key considerations. First, it effectively resolves dependency issues in network data, enabling more accurate relationship analysis. Second, it enhances reliability by statistically validating relationships against randomized matrices. Third, it is widely applicable to various network studies, including social network and activity pattern analysis, and can be integrated with traditional correlation methods such as Pearson and Spearman correlation analyses, facilitating a more comprehensive interpretation of results [27].

## Results

### Demographic characteristics

The analysis results of the demographic characteristics of the study participants are presented in Table 2 below.

### Analysis of network distribution characteristics

The results of the analysis of network distribution characteristics are presented in Table 3 below. A high density within the leisure activity network indicates a significant number of connections between leisure activities, suggesting that individuals engage in a variety of leisure activities. high-SWB group demonstrate a greater number of connections between leisure activities compared to those with low-SWB group, with values of 0.616 on weekdays and 0.966 on weekends. This indicates that higher-SWB group engage in a wider range of leisure activities.

Furthermore, a high level of inclusiveness signifies that there are fewer leisure activities that are not performed or are performed only minimally, which implies a larger number of specific leisure activities undertaken throughout the day. As a result, high-SWB group exhibit a higher number of specific leisure activities, with scores of 0.939 on weekdays and 1.000

**Table 2. Results of general characteristics (N = 482).**

| Classification | | High-SWB group (n = 241) | Low-SWB group (n = 241) |
|---|---|---|---|
| **Age(average)** | | 15.02 | 15.50 |
| **Sex** | Male | 110 | 107 |
| | Female | 131 | 134 |
| **Education Level** | Elementary | 40 | 40 |
| | Middle | 63 | 57 |
| | High | 138 | 144 |
| **Leisure Satisfaction** | Very Satisfied | 33 | 21 |
| | Satisfied | 45 | 54 |
| | Neutral | 77 | 81 |
| | Dissatisfied | 60 | 60 |
| | Very Dissatisfied | 26 | 25 |
| **Health Status** | Very Good | 79 | – |
| | Good | 162 | – |
| | Neutral | – | 153 |
| | Poor | – | 75 |
| | Very Poor | – | 13 |

on weekends, compared to their peers with low-SWB group. This suggests that there are fewer isolated leisure activities among higher-SWB group, enabling them to participate in a greater number of leisure activities.

A higher average distance within the leisure activity network indicates that connections between leisure activities become more challenging, leading to a reduction in the number of leisure activities performed simultaneously throughout the day. However, high-SWB group show a lower average distance, with values of 1.300 on weekdays and 1.304 on weekends, compared to those with lower-SWB group. This suggests that connections between leisure activities can be established more easily within a 24-hour period, allowing high-SWB group to engage in more simultaneous leisure activities than those with lower-SWB group.

## Centrality analysis

**Degree centrality. Weekdays:** The analysis results regarding connection centrality are presented in Table 4 and Figs 2 and 3. Table 4 summarizes only the top 10 activity items. Examining the most significant leisure activities of high-SWB group over a 24-hour period on weekdays, the results are as follows: 1st place is Sleeping (0.93), jointly 2nd place

**Table 3. Results of network distribution property.**

| Participant | | Density | Inclusiveness | Mean distance | Isolated nodes |
|---|---|---|---|---|---|
| **Weekday** | **High-SWB group** | 0.616 | 0.939 | 1.300 | 3 |
| | **Low-SWB group** | 0.275 | 0.653 | 1.349 | 17 |
| **Weekend** | **High-SWB group** | 0.966 | 1.000 | 1.304 | 0 |
| | **Low-SWB group** | 0.270 | 0.673 | 1.400 | 16 |

includes Face-to-Face Socializing, Video/Audio Calls, Community Participation, and Watching Video (0.91), and jointly 6th place includes Mobile Games, Doing Nothing and Resting, and Travel Related to Cultural and Leisure Activities (0.89).

For low-SWB group, the results show a tie for 1st place between Sleeping and Face-to-Face Socializing (0.64), 3rd place is Watching Live TV (0.60), 4th place is Purchasing Goods Offline (0.58), 5th place is Watching Video (0.56), jointly 6th place includes Community Participation, Searching the Internet, and Doing Nothing and Resting (0.54), and jointly 9th place includes Video/Audio Calls and Group Games/Play (0.52).

**Weekends:** The analysis results regarding connection centrality are presented in Table 5 and Figs 4 and 5. Table 5 summarizes only the top 10 activity items. Examining the most significant leisure activities of high-SWB group over a 24-hour period on weekends, the results are as follows: jointly 1st place includes Sleeping, Sleeplessness, Pet Care, Purchasing Goods Offline, Face-to-Face Socializing, Video/Audio Calls, Text Messages or E-mails, Socializing via Social Networking Services, Community Participation, and Attendance at Religious Meetings/Gatherings (1).

For low-SWB group, the results show 1st place as Sleeping (0.66), jointly 2nd place includes Community Participation and Watching Live TV (0.60), 4th place is Watching Video (0.58), jointly 5th place includes Purchasing Goods Offline, Text Messages or E-mails, Reading Books, and Mobile Games (0.56), and jointly 9th place includes Face-to-Face Socializing and Socializing via Social Networking Services (0.54).

## Cohesive structure analysis results

**Cohesive structure of weekday leisure networks.** The results of analyzing the cohesive structure of weekday leisure activity networks among high and low-SWB group are presented in Table 6 and Figs 6 and 7 below. Thicker lines indicate stronger relational ties between nodes, whereas thinner lines signify weaker relational ties. Additionally, nodes with connecting lines between groups serve as intermediary nodes, facilitating interactions across groups.

The analysis of cohesive structure followed a three-step process: in the first stage, activity items from the time-use survey data were categorized into six leisure activities based on the South Korea's Leisure Life Time Survey by Statistics Korea and were color-coded accordingly. In the second stage, connections within cohesive groups of leisure activities were analyzed. Finally, in the third stage, three researchers collaboratively named the themes of each cohesive group.

The analysis of the weekday leisure activity network cohesion among the high-SWB group revealed three subgroups, with themes characterized as follows: Group G1, "Social Participation-Related Leisure"; Group G2, "Individual Leisure in Cultural Activities"; and Group G3, "Restorative Leisure Related to Sleep and Relaxation." This structure indicates that high-SWB adolescents tend to engage in a balanced mix of social, cultural, and restorative activities, with a strong emphasis on interpersonal interactions and diverse leisure engagement.

**Table 4. Results of weekday degree centrality.**

| Rank | High-SWB group | Index | Rank | Low-SWB group | Index |
|---|---|---|---|---|---|
| 1 | Sleeping | 0.93 | 1 | Sleeping | 0.64 |
| 2 | Face-to-face socializing | 0.91 | 1 | Face-to-face socializing | 0.64 |
| 2 | Video/audio calls | 0.91 | 3 | Watching live TV | 0.60 |
| 2 | Community participation | 0.91 | 4 | Purchasing goods offline | 0.58 |
| 2 | Watching video | 0.91 | 5 | Watching video | 0.56 |
| 6 | Mobile games | 0.89 | 6 | Community participation | 0.54 |
| 6 | Doing nothing and resting | 0.89 | 6 | Searching the Internet | 0.54 |
| 6 | Travel related to cultural and leisure activities | 0.89 | 6 | Doing nothing and resting | 0.54 |
| 9 | Watching live TV | 0.87 | 9 | Video/audio calls | 0.52 |
| 9 | PC games | 0.87 | 9 | Group games/play | 0.52 |

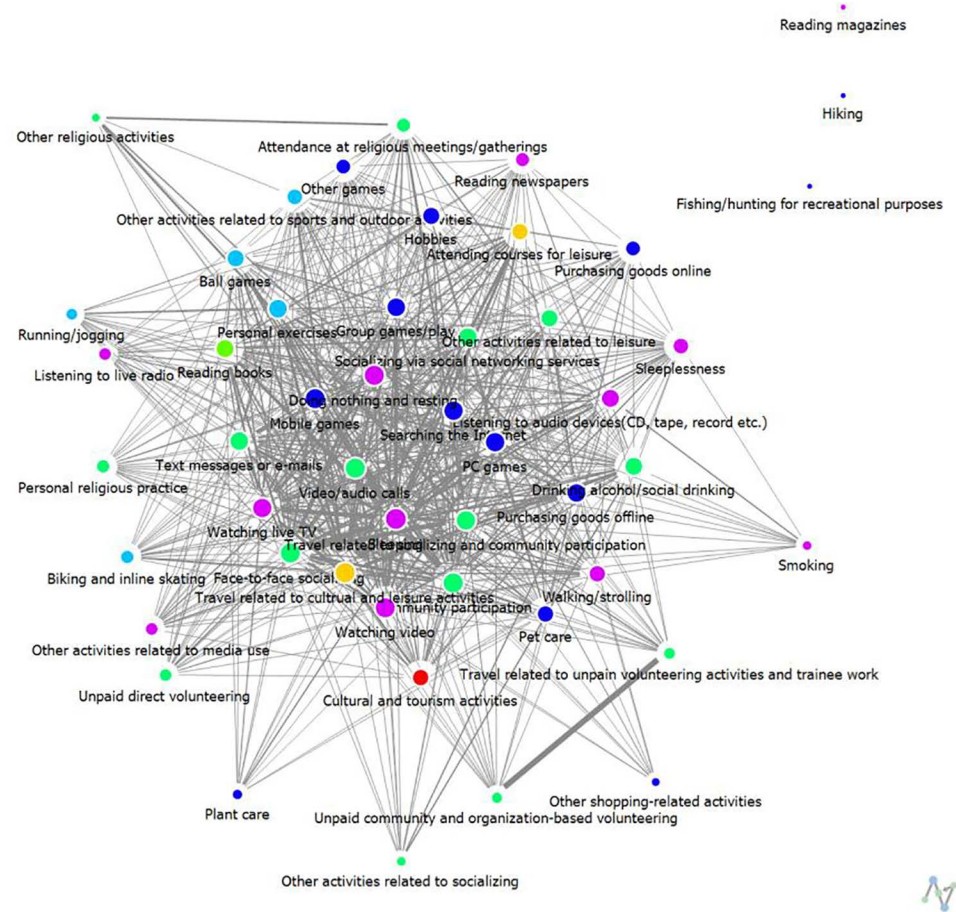

**Fig 2. The results of degree centrality of weekday leisure activities among High-SWB group.**

In contrast, the analysis of the weekday leisure activity network cohesion among the low-SWB group identified seven subgroups, each with the following thematic tendencies: Group G1, "Individual Hobbies and Leisure"; Group G2, "Group-Related Active Leisure"; Group G3, "Participatory Leisure Related to Relaxation"; Group G4, "Social Interaction-Related Participatory Leisure"; Group G5, "Community-Oriented Cultural Leisure"; Group G6, "Individual Leisure Related to Media"; and Group G7, "Sleep-Related Individual Leisure."

Notably, the low-SWB group exhibits a more fragmented network structure, with a greater number of subgroups, suggesting a lack of cohesion in their leisure activities. Furthermore, their engagement in social participation-related leisure appears less prominent, as activities such as social networking, online content consumption, and gaming dominate their network (G6: "Individual Leisure Related to Media"). This trend highlights that low-SWB adolescents rely more on digital interactions rather than in-person social engagement during weekdays. In contrast, the high-SWB group demonstrates a more integrated and socially connected network, where face-to-face interactions and community participation play a central role. This distinction underscores the importance of fostering real-world social connections and diverse leisure engagement to enhance subjective well-being among adolescents.

**Cohesive structure of weekend leisure networks.** The cohesive structure analysis results for the weekend leisure activity networks of high and low-SWB group levels are presented in Table 7 and Figs 8 and 9. The results of this study

   

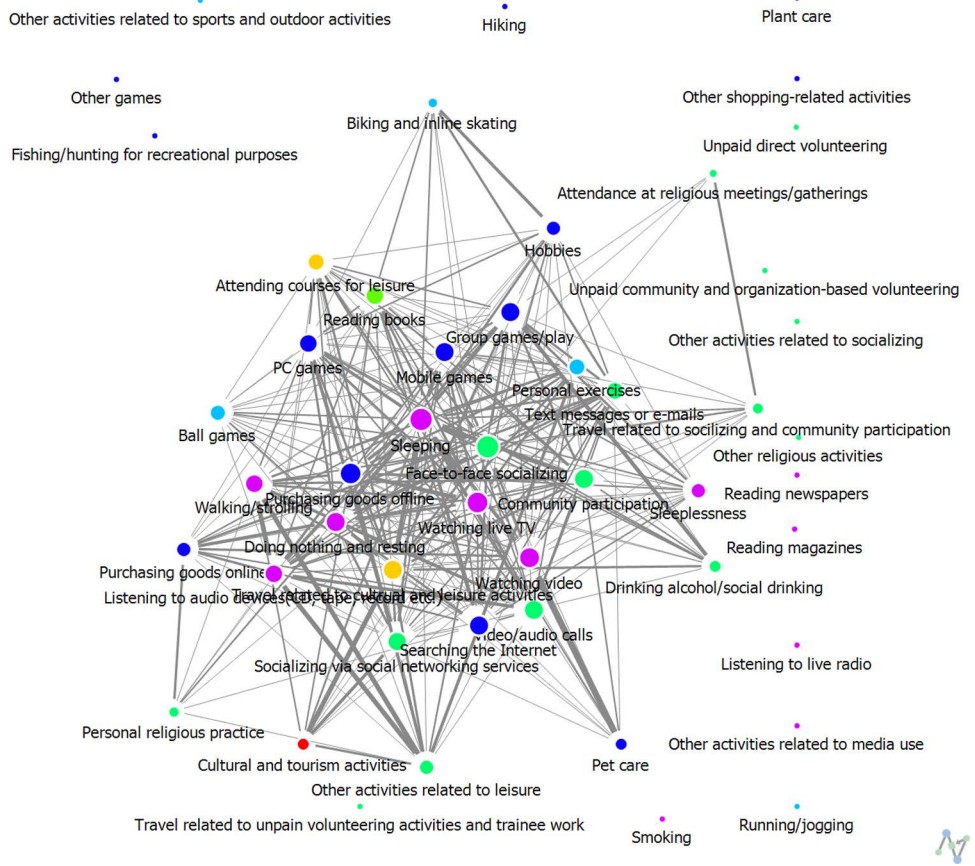

**Fig 3. The results of degree centrality of weekday leisure activities among Low-SWB group.**

identified six major subgroups among adolescents with high levels of subjective well-being (SWB), each characterized by distinct thematic attributes. First, Group G1 was classified as "Active Leisure Related to Gaming," encompassing activities involving active participation in digital games and interactive entertainment. Second, Group G2 was categorized as "Group-Related Active Leisure," characterized by community-based leisure activities that involve physical engagement.

**Table 5. Results of weekend degree centrality.**

| Rank | High-SWB group | Index | Rank | Low-SWB group | Index |
|---|---|---|---|---|---|
| 1 | Sleeping | 1 | 1 | Sleeping | 0.66 |
| 1 | Sleeplessness | 1 | 2 | Community participation | 0.60 |
| 1 | Pet care | 1 | 2 | Watching live TV | 0.60 |
| 1 | Purchasing goods offline | 1 | 4 | Watching video | 0.58 |
| 1 | Face-to-face socializing | 1 | 5 | Purchasing goods offline | 0.56 |
| 1 | Video/audio calls | 1 | 5 | Text messages or e-mails | 0.56 |
| 1 | Text messages or e-mails | 1 | 5 | Reading books | 0.56 |
| 1 | Socializing via social networking services | 1 | 5 | Mobile games | 0.56 |
| 1 | Community participation | 1 | 9 | Face-to-face socializing | 0.54 |
| 1 | Attendance at religious meetings/gatherings | 1 | 9 | Socializing via social networking services | 0.54 |

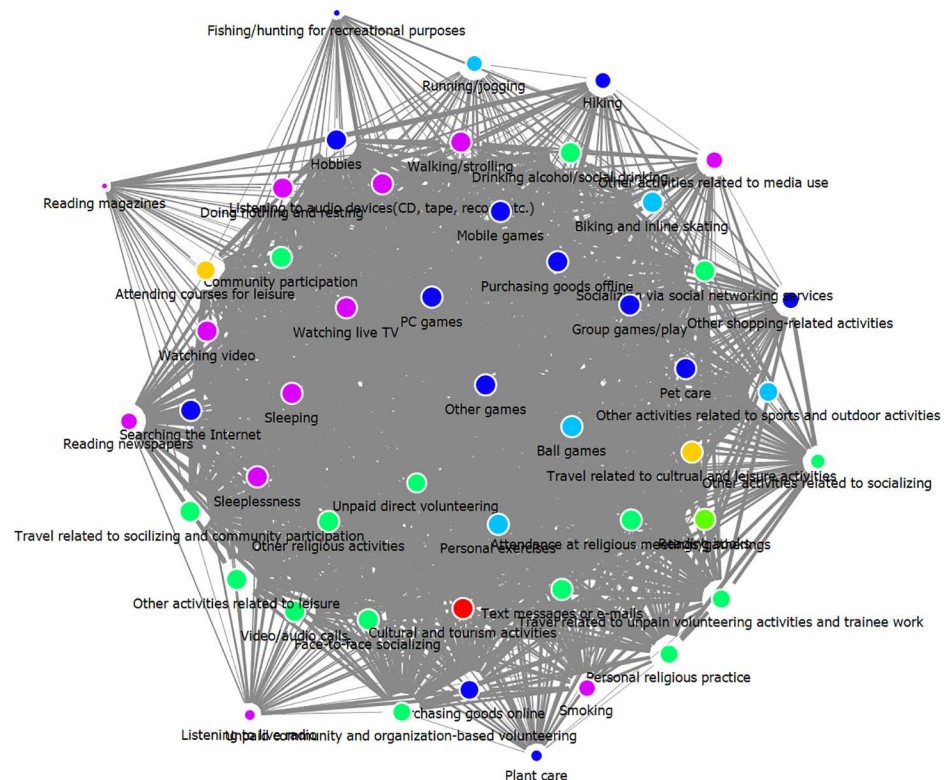

**Fig 4. The results of degree centrality of weekend leisure activities among High-SWB group.**

Third, Group G3 was defined as "Active Leisure Related to Cultural Activities," centering on participation in cultural domains such as arts, music, and creative activities. Fourth, Group G4 was labeled as "Passive Leisure Related to Media," comprising predominantly consumption-oriented activities such as watching videos, listening to music, and browsing online content. Fifth, Group G5 was identified as "Active Leisure Related to Relaxation," consisting mainly of physical and mental recovery activities such as walking and light exercise. Lastly, Group G6 was categorized as "Socially Engaged Leisure Related to Sleep," which primarily revolved around sleep but also included social interactions such as late-night conversations and relaxed engagements with friends.

Conversely, adolescents with low SWB exhibited a more fragmented structure, with seven distinct subgroups identified, reflecting a more restricted and individualized pattern of leisure activities. First, Group G1 was classified as "Participatory Leisure Related to Online Activities," characterized by engagement in virtual interactions and digital-based entertainment. Second, Group G2 was categorized as "Individual Leisure Related to Relaxation," comprising predominantly solitary leisure activities focused on rest. Third, Group G3 was identified as "Participatory Leisure Related to In-Person Social Interaction," which included face-to-face social interactions; however, compared to their high-SWB counterparts, the scope and frequency of such activities were relatively limited. Fourth, Group G4 was labeled as "Participatory Leisure Related to Media," centering on content consumption, which demonstrated a more passive nature compared to media-related activities in high-SWB groups. Fifth, Group G5 was defined as "Participatory Leisure Related to Mobile Activities," characterized by mobile-based activities such as texting and social media engagement. Sixth, Group G6 was classified as "Non-Active

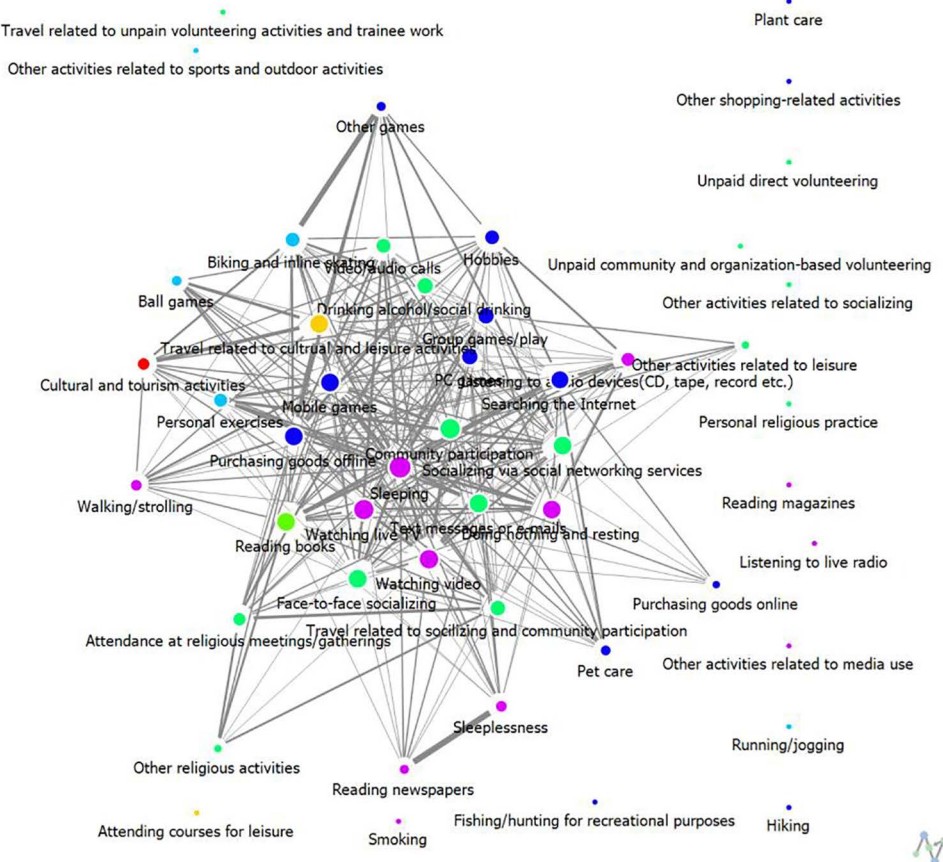

**Fig 5. The results of degree centrality of weekend leisure activities among Low-SWB group.**

Leisure Related to Sleep," emphasizing sleep-centric leisure with minimal active participation. Lastly, Group G7 was identified as "Participatory Leisure Related to Offline Activities," yet, in comparison to similar activities observed in high-SWB groups, the level of social connectivity within these networks appeared weaker.

These findings suggest that adolescents with high SWB tend to engage in a diverse range of leisure activities, balancing both active participation and social interactions, whereas those with low SWB exhibit a more fragmented leisure network, with a stronger inclination toward online-based, individual, and passive activities. Notably, low-SWB groups demonstrated a relative deficiency in face-to-face social interactions, with a predominant focus on online and solitary activities, highlighting the potential role of social engagement as a crucial factor in the relationship between SWB and leisure activities. The implications of this study emphasize the importance of leisure activity patterns and social relationships in fostering adolescent well-being, underscoring the need for policy and intervention strategies that enhance social connectivity and promote a more balanced engagement in leisure activities among low-SWB groups.

### QAP correlation analysis results

The results of the Quadratic Assignment Procedure (QAP) correlation analysis are shown in Table 8 below. The QAP correlation analysis examines the statistical significance between two networks using a non-parametric test with permutation. The correlation coefficients were found to be 0.55 ($p < 0.01$) between high-SWB group on weekdays and those with

**Table 6. Results of weekday cohesion structure analysis.**

| Participants | Group | Theme | Subgroup |
|---|---|---|---|
| **High-SWB group** | G1 | Social Participation-Related Leisure | Face-to-face socializing, Travel related to socilizing and community participation, Other activities related to leisure |
| | G2 | Individual Leisure in Cultural Activities | Travel related to cultural and leisure activities, Personal exercises, Drinking alcohol/social drinking, Cultural and tourism activities |
| | G3 | Restorative Leisure Related to Sleep and Relaxation | Sleeping, Reading books, Community participation, Socializing via social networking services, Text messages or e-mails, Video/audio calls, Purchasing goods online, Purchasing goods offline, Pet care, Watching live TV, Sleeplessness, Walking/strolling, Doing nothing and resting, Watching video, Listening to audio devices (CD, tape, record etc.), Mobile games, PC games, Hobbies, Ball games, Group games/play, Searching the Internet |
| **Low-SWB group** | G1 | Individual Hobbies and Leisure | Hobbies, Biking and inline skating |
| | G2 | Group-Related Active Leisure | Group games/play, Drinking alcohol/social drinking, Sleeplessness |
| | G3 | Participatory Leisure Related to Relaxation | Doing nothing and resting, Purchasing goods online, Personal religious practice |
| | G4 | Social Interaction-Related Participatory Leisure | Face-to-face socializing, Travel related to socializing and community participation, Attendance at religious meetings/gatherings |
| | G5 | Community-Oriented Cultural Leisure | Travel related to cultural and leisure activities, Attending courses for leisure, Cultural and tourism activities, Socializing via social networking services |
| | G6 | Individual Leisure Related to Media | Watching live TV, Reading books, Purchasing goods offline, Other activities related to leisure, Personal exercises, Walking/strolling, Listening to audio devices (CD, tape, record etc.), Video/audio calls |
| | G7 | Sleep-Related Individual Leisure | Sleeping, Text messages or e-mails, Community participation, Ball games, Searching the Internet, PC games, Mobile games, Watching video, Pet care |

low-SWB on weekends, 0.671 (p < 0.01) between low-SWB on weekdays, and 0.625 (p < 0.01) between those with low-SWB on weekends. These results indicate that the leisure activity networks are statistically significant in relation to each other.

## Discussion

This study analyzed the structural differences in adolescent leisure activity networks based on subjective well-being (SWB) levels and found that while adolescents in the high-SWB group actively engaged in a diverse range of leisure activities throughout both weekdays and weekends, maintaining strong network connectivity through various forms of social interaction such as in-person meetings, video and audio calls, text and email exchanges, and social media interactions, those in the low-SWB group exhibited a more restricted range of activities, predominantly engaging in sedentary or solitary pursuits with lower network cohesion and connectivity. These differences can be attributed to the high-SWB group's greater propensity for forming open social relationships and exploring new activities, resulting in higher network density, greater inclusiveness, shorter mean distance, and fewer isolated nodes, facilitating a more integrated and interconnected activity network.

Conversely, the low-SWB group exhibited more limited social engagement and a stronger dependency on specific activities, leading to lower network density, reduced inclusiveness, longer mean distance, and a higher prevalence of isolated nodes, thereby forming a more fragmented network structure. Notably, screen-based activities functioned differently depending on SWB levels, serving as a tool for enhancing social participation in the high-SWB group, while reinforcing isolated behavioral patterns in the low-SWB group. These findings underscore the critical role of SWB in shaping not only the type and frequency of adolescent leisure activities but also the structural properties and connectivity of their leisure activity networks, with significant implications for promoting more balanced and socially integrative leisure engagement.

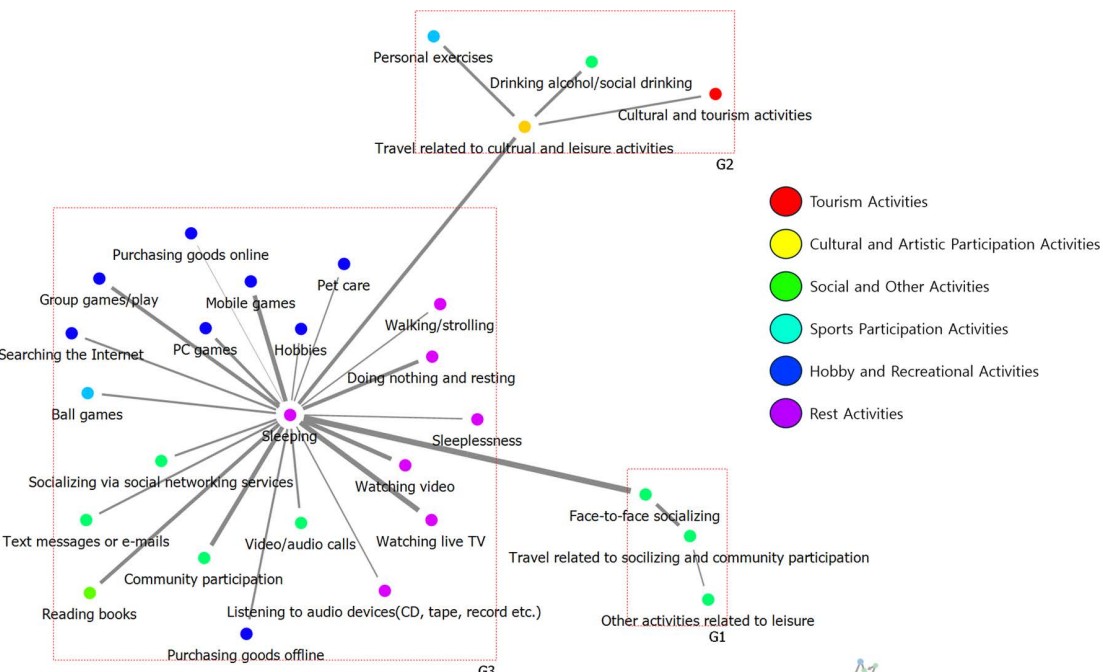

**Fig 6. The cohesive structure of weekday leisure activities among High-SWB group.**

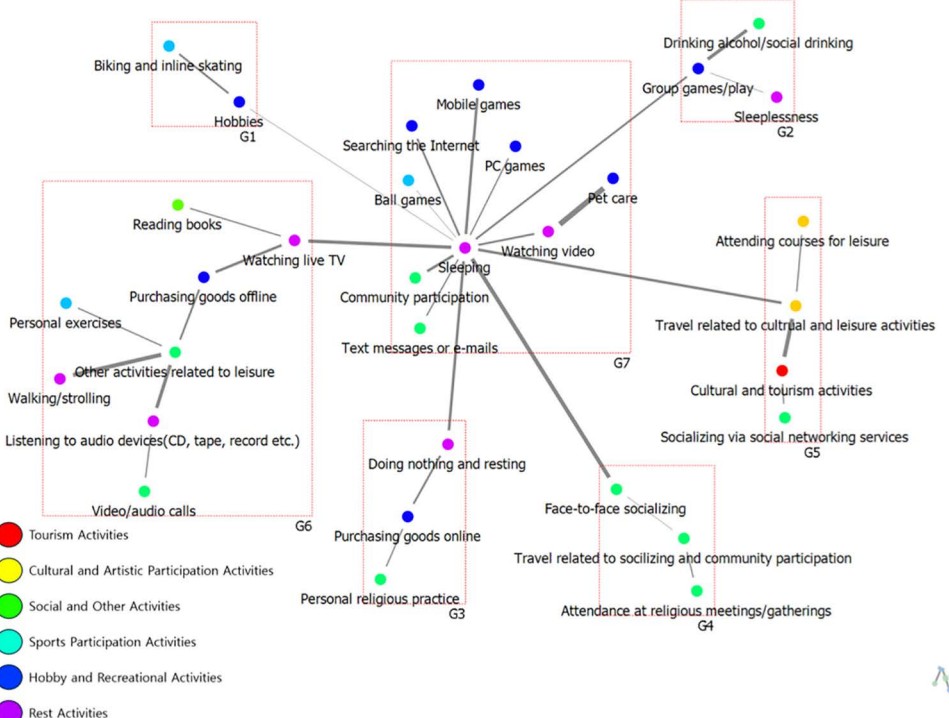

**Fig 7. The cohesive structure of weekday leisure activities among Low-SWB group.**

**Table 7. Results of weekend cohesion structure analysis.**

| Participants | Group | Theme | Subgroup |
|---|---|---|---|
| High-SWB group | G1 | Active Leisure Related to Gaming | Mobile games, Ball games, Travel related to cultural and leisure activities |
| | G2 | Group-Related Active Leisure | Group games/play, Reading books, Purchasing goods offline, Drinking alcohol/social drinking |
| | G3 | Active Leisure Related to Cultural Activities | Cultural and tourism activities, Other activities related to leisure, Sleeplessness, Walking/strolling, Travel related to socializing and community participation |
| | G4 | Passive Leisure Related to Media | Watching live TV, Watching video, Hobbies, Searching the internet, Pet care |
| | G5 | Active Leisure Related to Relaxation | Doing nothing and resting, Text messages or e-mails, Personal exercises, Community participation, Video/audio calls, Purchasing goods online |
| | G6 | Socially Engaged Leisure Related to Sleep | Sleeping, PC games, Listening to audio devices (CD, tape, record etc.), Face-to-face socializing, Socializing via social networking services |
| Low-SWB group | G1 | Participatory Leisure Related to Online Activities | Searching the internet, Pet care, Purchasing goods online |
| | G2 | Individual Leisure Related to Relaxation | Doing nothing and resting, Listening to audio devices (CD, tape, record etc.) |
| | G3 | Participatory Leisure Related to In-Person Social Interaction | Face-to-face socializing, Travel related to socializing and community participation, Attendance at religious meetings/gatherings |
| | G4 | Participatory Leisure Related to Media | Watching live TV, Group games/play, Other activities related to leisure |
| | G5 | Participatory Leisure Related to Mobile Activities | Mobile games, Walking/strolling, Personal exercises, Socializing via social networking services |
| | G6 | Non-Active Leisure Related to Sleep | Sleeping, Watching video, Community participation, Reading books, Sleeplessness, Text messages or e-mails, PC games |
| | G7 | Participatory Leisure Related to Offline Activities | Purchasing goods offline, Video/audio calls, Drinking alcohol/social drinking, Travel related to cultural and leisure activities, Ball games, Cultural and tourism activities |

Our findings align with prior research, which indicates that high-SWB group tend to engage in more diverse and socially interactive leisure activities [28]. Additionally, as highlighted in previous studies [29], adolescents' primary leisure activity involves social interactions with friends; this was similarly observed in our study, where higher-SWB group actively engaged in social activities. [11] noted that adolescents tend to invest more time in social activities as they grow, while [28] emphasized that meeting friends is central to adolescent leisure activities, which further supports the relationship between subjective well-being and social interaction found in this study. The observation that in-person and social network interactions are more frequent among low-SWB group may be due to their tendency to form connections with similar-minded peers within existing social networks, rather than establishing broader interpersonal relationships.

This finding suggests that school-based interventions, such as friend-making and job-experience programs, could be beneficial in promoting social participation for lower-SWB group. Both the high-SWB and low-SWB groups spent significant time on screen-based activities (e.g., video watching, gaming, live streaming, and internet browsing), consistent with prior studies [30,31]. However, their usage patterns differed. The high-SWB group balanced screen time with academics and social interactions, using online engagement as a tool for social connectivity. In contrast, the low-SWB group, often facing health or social challenges, relied more on screen-based activities due to limited academic and outdoor engagement. Research [32,33] suggests that low-SWB adolescents prefer sedentary leisure and report higher satisfaction with non-physical activities, partly due to social stigma. These findings indicate that screen-based activities are central to adolescent leisure but serve different functions: as social tools for high-SWB adolescents and as solitary, consumptive activities for those with low SWB, aligning with existing literature.

This study found that both high- and low-SWB groups showed limited engagement in sports-related activities, ranking individual exercise 15th and 19th, respectively. This suggests a general lack of interest in sports among adolescents, consistent with research indicating a shift toward sedentary leisure preferences [32]. Some adolescents may also prefer

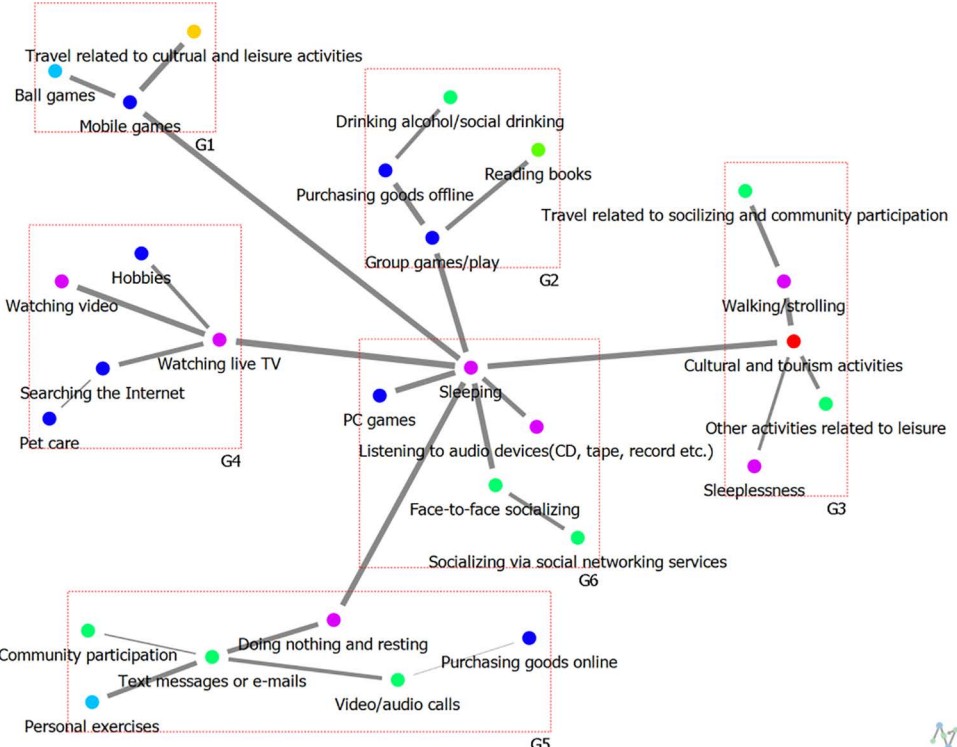

**Fig 8. The cohesive structure of weekend leisure activities among High-SWB group.**

solitary, non-physical activities due to individual choice rather than social or psychological factors [31,33]. While this study does not attribute low sports participation to a single cause, promoting balanced leisure engagement remains important for adolescent health and well-being.

Furthermore, high-SWB group displayed a balanced participation in various leisure activities over the weekend, with a centrality index of 1 across all activities. In contrast, lower-SWB group were more focused on specific activities, with notable differences in indices across leisure activities, indicating restricted leisure activity options. Cohesive structure analysis revealed that high-SWB group had closely interconnected leisure activity networks, engaging in various types of leisure activities such as socializing, cultural activities, sleep, and relaxation in an interconnected manner. Especially on weekends, activities like gaming, group activities, and cultural activities formed a highly cohesive leisure network, suggesting that high-SWB group make more varied use of their leisure time.

Conversely, lower-SWB group showed limited connectivity within their networks, focusing on a few activities like hobbies, socializing, and sleep during the weekdays, and online interaction, media use, and other specific activities over the weekends. Consequently, their leisure networks were less cohesive, with fewer connections between activities and a lack of diversity. This contrast in cohesion highlights how leisure activity networks are structured differently according to subjective well-being levels, with high-SWB group displaying more cohesive and interconnected networks, while those with low well-being showed limited and less cohesive networks. This underscores the importance of supporting adolescents in balancing diverse leisure activities. According to [34], cohesive structure analysis allows for the identification of characteristic subgroups within a network, and structural features of sub-networks between specific nodes can be analyzed.

Unlike traditional leisure studies, this research holds significance in its application of network analysis to visually capture the relationships and connectivity among leisure activities based on adolescents' subjective well-being levels.

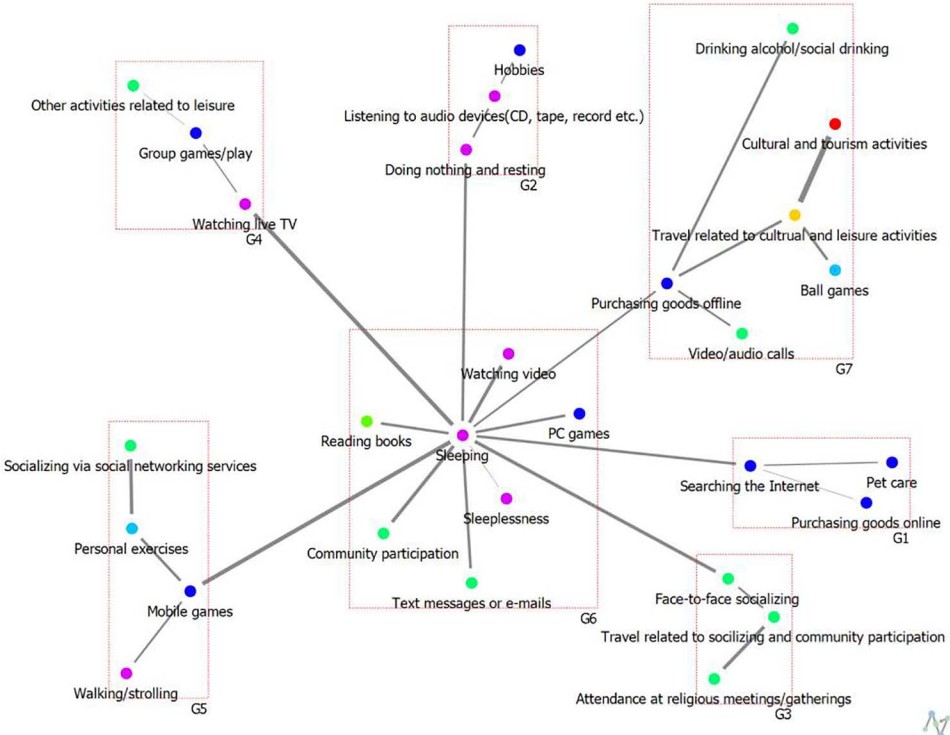

**Fig 9. The cohesive structure of weekend leisure activities among Low-SWB group.**

Traditional leisure studies have primarily relied on quantitative statistical analyses to numerically assess the frequency, duration, and various factors related to leisure activities using large-scale samples. While this methodology is useful for deriving statistical trends from the data, it has limitations in deeply understanding the relationships and interactions among individual activities. In contrast, this study applies Social Network Analysis (SNA) to visually capture the relationships and connectivity among leisure activities based on adolescents' subjective well-being levels.

This study empirically confirms that higher levels of subjective well-being (SWB) among adolescents are associated with increased connectivity and cohesiveness within their leisure activity networks, as evidenced through the analysis of network distribution characteristics, centrality, and cohesive structure. High-SWB group tend to engage in a diverse range of leisure activities in a balanced manner, forming multilayered social networks. In contrast, adolescents with low-SWB group exhibit a tendency to concentrate on specific activities, leading to a more restricted scope and limited connectivity within their leisure networks. These findings suggest that adolescent leisure activities extend beyond personal pastimes and play a crucial role in shaping social relationships and influencing psychological well-being.

**Table 8. Results of QAP correlation analysis.**

|  | High-SWB group on weekdays | High-SWB group on weekends | Low-SWB group on weekdays |
|---|---|---|---|
| **High-SWB group on weekends** | 0.55** | — | — |
| **Low-SWB group on weekdays** | 0.671** | 0.452** | — |
| **Low-SWB group on weekends** | 0.625** | 0.452** | 0.522** |

** $p < 0.05$

The results of the Quadratic Assignment Procedure (QAP) correlation analysis in this study empirically demonstrate the structural differences in leisure activity networks between weekdays and weekends according to the level of subjective well-being (SWB) among adolescents, providing significant implications. In particular, the findings indicate that low-SWB group tend to maintain similar leisure activity patterns throughout both weekdays and weekends, whereas those with high-SWB group exhibit a tendency to diversify their activities and develop a more balanced leisure activity network, particularly over the weekend.

Notably, the high correlation coefficients observed between the leisure activity networks of low-SWB group on weekdays and weekends (0.671, 0.625) suggest a strong inclination towards the repetition of familiar activities rather than the exploration of new ones. This observation aligns with existing literature, which suggests that lower -SWB group are more likely to adopt a restrictive leisure activity pattern due to social stigma and psychological barriers [32,33]. In contrast, the statistically significant correlation (0.55) between the weekday leisure activity network of high-SWB group and the weekend network of low-SWB group indicates that the scope of leisure activities undertaken by the latter during weekends remains at a level comparable to that of the former on weekdays. This finding underscores a distinct contrast in leisure engagement strategies, wherein high-SWB group actively expand their social and cultural activities over the weekend, whereas low-SWB group exhibit a constrained pattern of engagement, reinforcing the necessity for policy interventions aimed at facilitating greater leisure activity diversification and social integration.

Accordingly, it is essential to promote balanced leisure activities that incorporate both active participation and social interaction when formulating adolescent policies and intervention strategies. Specifically, structured community-based programs that enhance face-to-face interactions should be expanded, and targeted support should be provided to help low-SWB group establish stronger social networks. Additionally, strategies should be implemented to reduce excessive reliance on online and passive leisure activities. Furthermore, tailored interventions should be designed to not only support individual relaxation and well-being but also facilitate meaningful social connections. Such an approach is expected to enhance adolescents' mental health and well-being while mitigating the negative effects of social isolation.

## Limitations

This study has limitations in fully reflecting the diverse characteristics within the adolescent group, as it does not account for general demographic variables such as gender, age, and grade level. Future research should further refine demographic segmentation to conduct a more in-depth analysis of how patterns of leisure activity participation vary by gender and age group.

Furthermore, as this study is based on data from a specific time point in 2019, there is a need for longitudinal analyses to examine changes in adolescent leisure activities over time and explore the temporal relationship between subjective well-being and leisure participation. Such research would provide a more robust foundation for designing effective leisure intervention programs and informing policy improvements that consider adolescents' subjective well-being.

Additionally, the methodological framework employed in this study, including weighted networks, bipartite networks, and multilayer networks, requires further clarification to ensure a more precise understanding of the analytical approaches. The scope of the analysis was also somewhat limited; incorporating additional network metrics, such as centrality and modularity, could enhance the depth and comprehensiveness of the findings.

Moreover, improving the precision of weekday-versus-weekend comparisons would contribute to the reliability of multilayer network analysis. While maintaining comparability with previous research, integrating advanced network visualization techniques and analytical methods could provide deeper insights into the structural dynamics of adolescent leisure activity networks.

## Conclusion

The academic significance of this study lies in its use of the 2019 Time Use Survey data from Statistics Korea and its application of social network analysis to examine the relationships within adolescent leisure activities based on subjective

well-being. By constructing leisure activity networks, this study aims to visually explore meaningful and purposeful leisure activities. Comparing leisure activity networks of adolescents with high and low subjective well-being allowed for a multi-faceted view of adolescent leisure relationships beyond traditional statistical analysis, identifying leisure activities that are more central to daily life than mere frequency suggests. This study contributes to addressing mental health challenges related to declining subjective well-being among adolescents by providing insights for policy formulation and legislative amendments related to youth leisure activities. Furthermore, we propose that social network analysis be utilized in future research on adolescent leisure activity differences.

## Acknowledgments

We would like to express my gratitude to Professor Sanghee Lee for the invaluable support provided throughout this research. Professor Lee generously shared insightful perspectives and offered tremendous support, which greatly contributed to the success of this study.

## Author contributions

**Conceptualization:** Jinseok Oh.

**Data curation:** Jinseok Oh.

**Methodology:** Jinseok Oh.

**Project administration:** Jin-Hyuck Park.

**Writing – original draft:** Jinseok Oh, Sungmin Son, Jin-Hyuck Park.

**Writing – review & editing:** Jinseok Oh, Sungmin Son, Jin-Hyuck Park.

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
