## [Decision Letter · Decision Letter 0]

27 Jan 2025

PONE-D-24-52060An Analysis of Adolescent Leisure Activity Structure Based on Subjective Well-being: Focusing on Social Network AnalysisPLOS ONE

Dear Dr. Park,

Thank you for submitting your manuscript to PLOS ONE. After careful consideration, we feel that it has merit but does not fully meet PLOS ONE’s publication criteria as it currently stands. Therefore, we invite you to submit a revised version of the manuscript that addresses the points raised during the review process.

**ACADEMIC EDITOR:**Dear authors,

Thank you for your submission and your patience.

Five expert reviewers have evaluated your investigation.

Please, respond to all of them and submit your revised manuscript and response letter indicating your changes and comments.

Thank you.

We look forward to receiving your revised manuscript.

Kind regards,

Javier Fagundo-Rivera, PhD

Academic Editor

PLOS ONE

Journal Requirements:

“This research was supported the Soonchunhyang University Research fund and the Ministry of Education of the Republic of Korea and the National Research Foundation of Korea (NRF-2023S1A5C2A03099545).”

**Additional Editor Comments:**

Dear authors,

Thank you for your submission and your patience.

Five expert reviewers have evaluated your investigation.

Please, respond to all of them and submit your revised manuscript and response letter indicating your changes and comments.

Thank you.

Reviewers' comments:

Reviewer's Responses to Questions

**Comments to the Author**

1. Is the manuscript technically sound, and do the data support the conclusions?

Reviewer #1: Partly

Reviewer #2: Yes

Reviewer #3: Yes

Reviewer #4: Partly

Reviewer #5: Partly

2. Has the statistical analysis been performed appropriately and rigorously? 

Reviewer #1: I Don't Know

Reviewer #2: Yes

Reviewer #3: I Don't Know

Reviewer #4: No

Reviewer #5: Yes

3. Have the authors made all data underlying the findings in their manuscript fully available?

Reviewer #1: Yes

Reviewer #2: Yes

Reviewer #3: Yes

Reviewer #4: Yes

Reviewer #5: Yes

4. Is the manuscript presented in an intelligible fashion and written in standard English?

Reviewer #1: Yes

Reviewer #2: Yes

Reviewer #3: Yes

Reviewer #4: Yes

Reviewer #5: Yes

5. Review Comments to the Author

**Reviewer #1: **

First of all, I'd like to thank the authors for submitting their work to PLOS ONE. In this manuscript, the authors apply Social Network Analysis (SNA) to unveil characteristics and patterns in the leisure activity networks of adolescents. The focus lies on analyzing data from adolescents with both high and low subjective well-being (SWB). All the data comes from a South Korean survey conducted in 2019.

While the authors use SNA to bring new insights into this problem, differentiating it from purely statistical analyses, I feel that their analysis falls flat, and there is untapped potential. Regretfully, as it stands, I cannot recommend the article for publication without at least undergoing a major revision. Below, I outline the major and minor issues with the work.

#Major Issues

My main concern is that, although potentially interesting, the analysis feels too shallow for this journal. Perhaps the data inherently limits the outcome and scope, which would be unfortunate if the research aims for high impact. However, I believe there is room for improvement.

If I understood correctly, the data from the KOSTAT Time Use Survey includes anonymized respondents reporting the time spent on each activity during weekdays (either a single day or five weekdays—this remains unclear) and the weekend. Moreover, the survey includes information about each respondent's well-being and health status, assessed via a Likert scale.

Specifically, it is stated: "To identify adolescents with low levels of subjective well-being, individuals were categorized based on their subjective well-being scores (Likert scale 5–7) and health status scores (Likert scale 3–5)."

In Table 2, the general characteristics of each group (low and high SWB) are presented. There is one metric, leisure satisfaction, which is not defined or mentioned in the main text, nor is it explicitly stated that it is part of the dataset. The authors should clarify and expand on this. Interestingly, leisure satisfaction appears similar across groups, except for the 'very satisfied' category, where the high SWB group has a clear advantage. Regarding health status, the high SWB group shows 'very good' or 'good' outcomes, while the low SWB group is restricted to 'neutral' to 'very poor,' which makes sense. However, it is unclear how the division into high and low SWB groups is determined—whether it involves leisure satisfaction and health status or arises from an independent survey response. Clarifying this is critical.

Additionally, if SWB is not binary (low/high) but assessed on a continuous scale, why not analyze results while accounting for that granularity? A binary approach simplifies the analysis, but a more fine-grained perspective might yield deeper insights.

Regarding methods, the authors use a simple network to represent leisure activity relationships. If the data accounts for the time spent (or fraction of time) on each activity rather than a binary yes/no, the analysis could benefit from alternative network representations:

- Weighted network: Nodes could be weighted by the (normalized) population engaging in an activity, and edges by the (normalized) number of individuals participating in both activities.

- Bipartite network: One layer could represent respondents and the other the full set of activities, with connections linking individuals to their activities. A weighted approach could also apply here.

- Multilayer/multiplex network: Layers could differentiate between weekdays and weekends or online versus offline activities, for example.

Additionally, the authors might consider supervised and unsupervised machine learning techniques to either forecast SWB based on activity features or explore emerging clusters of individuals. When comparing leisure activity structures (across SWB groups or weekday/weekend divides), a comparison against a "null model" of random activity networks would reveal whether the observed structures are meaningful.

I am not suggesting that the authors apply all these frameworks, but implementing at least some of them, along with a broader set of network metrics, could significantly enhance the analysis and provide a richer characterization of the problem. As it stands, I feel the potential of this data is underutilized.

Finally, I noticed on the KOSTAT website that the Time Use Survey has been conducted several times historically:

- Sep. 1999: First survey

- Sep. 2004: Second survey

- March & Sep. 2009: Third survey

- July, Sep., & Dec. 2014: Fourth survey

- July, Sep., & Dec. 2019: Fifth survey

Extending the analysis to previous survey rounds would provide longitudinal data, illustrating how patterns have evolved over time. With an established methodology and workflow, this extension should not be too demanding.

#Minor Issues

Below is a collection of minor issues for improving the manuscript, regardless of how the major issues are addressed:

- Use South Korea instead of "National" when referring to the survey, given the journal's global scope.

- Lines 113–127 contain overlapping content. Restating the second paragraph (lines 120–123) to avoid redundancy would improve flow.

- Specifying software company names seems unnecessary. If retained, consider hyperlinking to the respective webpages instead of listing the company and country in parentheses.

- The sentence "For this reason, ethical approval for the study" is incomplete.

- The description of analysis methods (e.g., NetMiner software) is repeated unnecessarily.

- When discussing network density (e.g., "a greater number of connections between leisure activities"), avoid using "number of connections." Since the metric is density, terms like proportion or share would be more precise.

Table 6 contains typos: "cultrual" → "cultural," "activitie" → "activities."

- The analysis of recreational activities (lines 337–349) concludes that lower SWB groups prioritize personal leisure more. However, both groups seem dominated by screen-based activities, with only minor differences (e.g., reading for low SWB). This should be reconsidered.

- To reduce verbosity, consider abbreviations like "high-SWB group" instead of "adolescents with high subjective well-being."

- Figures:

- Use vector formats to improve image quality.

- Figure 1 could clarify that steps 4 and 5 are simultaneous, not sequential.

- Figure 2 could be integrated into Figures 3 and 4 as a legend, rather than standing alone.

- Figures 3–6 might benefit from a multi-panel layout (e.g., 2x2 grid) for direct comparison of high/low SWB and weekday/weekend differences.

# Final Thoughts

This manuscript might have potential but requires significant revisions to fully realize its contributions. By addressing the aforementioned issues and expanding the analysis, the authors can produce a more impactful and robust study. I encourage the authors to explore these suggestions and further refine their work. Thank you for considering my feedback.

**Reviewer #2: **

Some limitations might be known to researchers before the study starts, while others might be discovered as the investigation progresses. Regardless of whether these constraints were anticipated or not, or whether they were the consequence of methodology or research design, they should be specifically identified and addressed in the discussion section, the final section of your report. Most journals now require you to list any potential limitations of your research, and many of them request that you add this "limitations section" at the very conclusion of your work.

**Reviewer #3: **

This paper analyzes how leisure activities affect the well-being of young people in South Korea. The authors argue that, compared to young people with high levels of well-being, those with low levels of well-being have different network structures.

However, the particularly important aspect of the research, namely the network construction and analysis, is insufficient. Therefore, I would like to request a major revision with detailed additions.

1. Please elaborate on how you constructed the network from the specific data in lines 142-160, including the methodology for creating links based on co-occurrence relationships (e.g., if at least one person engages in both leisure activities, an edge is created), the weighting of links, and how the frequency of leisure activities influenced the network construction.

2. Please explain how you converted the 2-mode network to a 1-mode network for lines 168-170. In addition, clarify the types of nodes in the 2-mode network and specify the criteria used to remove redundant parts.

3. Please review this definition of density for Table 1. In Table 1, the density is defined as “the ratio of nodes actually connected to the total number of nodes,” but in general, density represents the ratio of links to the possible number of links.

4. Please review the network visualizations in Figures 3-6. The results in Table 3 and the network visualizations appear inconsistent. Which part of the network is being visualized? For example, Table 3 mentions isolated nodes, but they are not present in the visualization. Additionally, the visualized networks do not appear to be dense networks with a mean distance of less than 1.4.

5. Please add more discussion about Figures 3 and 4. In lines 314-317, it is said that results indicated that adolescents with high subjective well-being exhibited distinct network characteristics such as density, inclusiveness, average distance, and isolated nodes and engaged in diverse leisure activities on both weekdays and weekends. However, in Figure 3, the most central node is sleeping in adolescents with high subjective well-being. Does it show the diverse leisure activities on weekdays?

6. Please clearly describe the discussion or interpretation of the results of Figures 5 and 6. From Figures 5 and 6, it is not possible to read that people with low well-being have less diversity in their social activities or that they have more online interactions.

7. Please discuss the results of the QAP test in more detail for line 311. The correlation between the leisure networks of young people with high well-being and those with low well-being is 0.671 on weekdays but drops to 0.452 on weekends, suggesting that the structure of leisure networks may differ more on weekends.

8. The discussion with academic work is a little difficult to understand. For lines 345-348, why do students with high subjective well-being concentrate on their studies, while those with low subjective well-being are limited in their approach to their studies? As Table 2 shows, there is indeed a difference in health between the two groups, but is this enough to have a direct impact on academics?

9. The discussion of sports is also a little difficult to understand. In lines 351-353, adolescents with high subjective well-being ranked individual exercise at 15th, while those with lower well-being ranked it at 19th, indicating that both groups engaged minimally in sports-related activities with low network centrality. If that is the case, the connections with previous studies mentioned in lines 353-356 seem overstated. All we can infer is that sports are generally not popular among young people.

**Reviewer #4: **

General Assessment:

This manuscript explores the relationship between adolescents' subjective well-being (SWB) and their leisure activity structures using Social Network Analysis (SNA). The study leverages data from the 2019 Time Use Survey of Statistics Korea, analyzing leisure activity networks for 241 adolescents with high SWB and 241 with low SWB. While the manuscript presents a novel approach to understanding the interplay between leisure activities and SWB, there are critical methodological and structural issues that need to be addressed to enhance its scientific rigor and clarity.

Comments:

-The research aims are clear but lack precise alignment with the findings.Ensure the stated research objectives explicitly correspond to the results. For example, elaborate on how the study’s insights contribute to policy formulation and interventions for adolescents.

-The methodology section is robust but requires additional detail for reproducibility and clarity.

-The manuscript mentions data extraction from the 2019 Time Use Survey. However, there is limited explanation of the ethical considerations and specific survey variables used. Include detailed descriptions of the extracted variables and their relevance to SWB and leisure activity analysis. Clearly state how the anonymized dataset ensures compliance with ethical standards.

-The use of SNA is innovative but inadequately explained. Provide a step-by-step description of how the 2-mode networks were converted into 1-mode networks. Clarify the criteria for defining connections between leisure activities and justify the selection of network metrics (e.g., density, inclusiveness, centrality).

-The classification of adolescents into high and low SWB groups based on Likert scale scores lacks justification. Provide a rationale for the chosen thresholds. Discuss whether these thresholds are supported by prior studies or validated through statistical methods.

-The manuscript briefly mentions the use of SPSS and NetMiner but does not detail how statistical significance was determined. Specify the statistical tests used to compare network metrics between groups and justify the significance level (e.g., p < 0.05).

-While the results are comprehensive, their interpretation is overly descriptive. Critically analyze the implications of differences in network metrics (e.g., higher density and inclusiveness for high SWB adolescents). Discuss the potential causal relationships and align findings with existing literature.

-Several figures and tables lack sufficient annotations and descriptions. Enhance the readability of figures (e.g., Figs 3-6) by including detailed legends and annotations. Ensure all tables have clear titles and explanations for each metric.

-The manuscript’s discussion of limitations is brief. Expand on the limitations, such as the reliance on a single dataset and potential cultural biases in the Korean context. Discuss how these factors might affect the generalizability of findings.

-The abstract is informative but overly dense. Simplify the language and highlight the key findings and implications in a more concise manner.

-The introduction provides useful context but could better integrate global literature on leisure activities and SWB. Include references to studies from diverse cultural settings to contextualize the findings.

-Terms like "cohesion analysis" and "centrality" are used without sufficient explanation for non-expert readers. Define technical terms upon first use to improve accessibility.

-The manuscript contains minor grammatical errors and awkward phrasing. Proofread the text to improve fluency and readability.

While the manuscript presents a novel approach to studying adolescent leisure activities and SWB, significant revisions are needed to enhance its methodological rigor, clarity, and overall impact. I look forward to reviewing a revised version that addresses these comments.

**Reviewer #5: **

Dear authors,

Thank you for your hard work in reaching this stage of your research. I truly appreciate your effort and recognize the importance of sharing these findings with the global readership. However, I have important concerns and suggestions that should be addressed prior to publication:

1. Could you clearly outline the theoretical and conceptual framework that guided your research? Including two paragraphs on this topic would enhance the clarity of your study. Additionally, it would be helpful to explain why SNA is the most suitable approach for your research.

2. What research questions are you addressing? What hypotheses are you testing? Every research study should aim to answer specific questions or test hypotheses. However, these elements are not explicitly stated in your manuscript and need to be clarified.

3. Your data was collected during a 2019 survey, which is now six years old. Could you address how your findings remain timely and relevant in the current context?

6. PLOS authors have the option to publish the peer review history of their article (what does this mean? ). If published, this will include your full peer review and any attached files.

**Do you want your identity to be public for this peer review?** For information about this choice, including consent withdrawal, please see our Privacy Policy .

Reviewer #1: No

Reviewer #2: No

Reviewer #3: **Yes: ** Masaki Chujyo

Reviewer #4: No

Reviewer #5: No

---

## [Author Response · Author response to Decision Letter 1]

11 Mar 2025

To the Editors and Reviewers of PLOS ONE,

We sincerely appreciate your review and valuable feedback on our study. We have carefully considered the comments from Reviewer and have incorporated several revisions to enhance the scientific rigor and clarity of our manuscript. Below, we provide our responses to the key points raised by the reviewer and outline our revisions.

[Reviewer 1]

<Major Issues>

1. The analysis in the study is too superficial, and data utilization is insufficient.

1) In Table 2, the group characteristics include an indicator called "leisure satisfaction," which has not been clearly defined and is not mentioned in the main text. It should be clarified whether leisure satisfaction was included in the study data.

- Thank you for your insightful comment. We have revised and supplemented the section related to the study data in the Methods section(line 130). The explanation regarding the analyzed data has been modified and clarified in the revised manuscript. (lines 206–213 on page 7).

- Thank you for your comment. We have defined the items leisure satisfaction and health status among the general characteristics used in this study and have added the relevant content to the revised manuscript. (lines 220–223 on page 7).

2) It remains unclear how the SWB groups were classified. A more detailed explanation is needed on whether both leisure satisfaction and health status were included or if the classification was based on independent survey responses.

- Thank you for your comment. We have revised and clarified the explanation regarding the classification method of the SWB groups in the Methods – Study Participants section(line 144) to ensure greater clarity. (lines 214–219 on page 7).

3) If SWB was assessed as a continuous variable rather than a dichotomous variable(high/low), it might be more appropriate to consider this continuity in the analysis.

- Dear Reviewer

Thank you very much for your valuable comments.

We acknowledge that evaluating and analyzing subjective well-being(SWB) as a continuous variable is a meaningful approach, particularly in consideration of data continuity. However, in the early stages of our study design, we adopted a structure that categorizes SWB into high and low groups to form the sample and answer the main research questions through group comparisons. We believe this design is appropriate for clearly analyzing differences between groups.

Incorporating SWB as a continuous variable would require fundamental modifications to the study design and analytical approach, which may compromise the consistency of our findings and the primary objectives of the study. Therefore, we kindly ask for your understanding that implementing the suggested approach in the current study is challenging.

Nevertheless, we will actively consider your suggestion in future research to analyze SWB as a continuous variable, aiming to derive more in-depth insights.

Sincerely

2. Expansion of Network Analysis Methods

1) The authors represent the relationships between leisure activities using a simple network. However, if the data includes not only participation(yes/no) but also the time spent on each activity, a more sophisticated network analysis could be conducted. Alternative network representations to consider include:

Weighted Network: Nodes can be weighted based on the (normalized) proportion of the population participating in a specific activity, while edges can be weighted based on the (normalized) number of individuals participating in both activities.

Bipartite Network: One layer can represent respondents, while the other layer includes all activities, modeling the connections between respondents and activities. Weights can also be applied to this structure.

Multilayer/Multiplex Network: Different layers can be used to separate weekday and weekend activities, or to distinguish between online and offline activities.

- Thank you for your in-depth feedback and valuable suggestions. We sincerely appreciate your deep interest in our study and your constructive advice. In response to the key issue raised regarding the representation of relationships between leisure activities as a simple network, we would like to explain our research approach and future directions for improvement.

(1) Response to the Comment on the Simplicity of Network Analysis

As the reviewer pointed out, the network model used in our study may appear to follow a simple dichotomous (yes/no) approach. However, this dataset is derived from secondary data collected through a time-use survey that examined 153 activities over 24 hours for the entire population of South Korea. In our study, we categorized 49 leisure activities for analysis.

When constructing the network, the source was set as individuals, the target as leisure activities, and the weight as the amount of time spent on each activity. Therefore, rather than being a simple network, our study incorporates characteristics of various network types.

(2) Data Structure and Analytical Possibilities

A. Weighted Network:

In this study, the weights reflect the amount of time each respondent spent on a specific activity, allowing for an analysis that considers the strength of connections rather than simply their presence or absence. Therefore, our study can be classified as a Weighted Network analysis.

B. Bipartite Network:

Our study constructs a 2-mode network(Bipartite Network) that represents the relationship between individuals and activities. In other words, nodes of the same type(e.g., activity-activity) are not directly connected; instead, the analysis focuses on the connections between individuals and activities. This distinguishes our approach from conventional simple network models.

C. Multilayer/Multiplex Network:

As analyzed in the Results section(starting from line 183), we compared leisure activities on weekdays and weekends, making this study applicable to Multilayer/Multiplex Network analysis. By separately comparing weekday and weekend activities, we were able to clearly analyze differences in leisure activity patterns.

(3) Future Research and Limitations

We intend to conduct follow-up studies incorporating the suggestions provided by the reviewer. Additionally, we would like to include several limitations, as outlined below. (lines 801–851 on page 22).

- Dear reviewer

This study has limitations in fully reflecting the diverse characteristics within the adolescent group, as it does not account for general demographic variables such as gender, age, and grade level. Future research should further refine demographic segmentation to conduct a more in-depth analysis of how patterns of leisure activity participation vary by gender and age group.

Furthermore, as this study is based on data from a specific time point in 2019, there is a need for longitudinal analyses to examine changes in adolescent leisure activities over time and explore the temporal relationship between subjective well-being and leisure participation. Such research would provide a more robust foundation for designing effective leisure intervention programs and informing policy improvements that consider adolescents’ subjective well-being.

Additionally, the methodological framework employed in this study, including weighted networks, bipartite networks, and multilayer networks, requires further clarification to ensure a more precise understanding of the analytical approaches. The scope of the analysis was also somewhat limited; incorporating additional network metrics, such as centrality and modularity, could enhance the depth and comprehensiveness of the findings.

Moreover, improving the precision of weekday-versus-weekend comparisons would contribute to the reliability of multilayer network analysis. While maintaining comparability with previous research, integrating advanced network visualization techniques and analytical methods could provide deeper insights into the structural dynamics of adolescent leisure activity networks.

2) After reviewing the website of Statistics Korea(KOSTAT), it appears that the Time Use Survey has been conducted multiple times. Expanding the analysis to include past survey data could provide insights into how adolescent leisure activity patterns have changed over time. Once the existing methodology is established, such an extension could be achieved with relatively little additional effort.

- Dear Reviewer

We sincerely appreciate your valuable feedback and constructive suggestions, as well as your deep interest in our research. The Time Use Survey has been conducted nationally every five years from 1999 to 2019. However, due to periodic changes in survey items, there are limitations in comparing all years. Additionally, while we aimed to compare the more recent 2014 and 2019 data, the 2014 survey did not include items measuring subjective well-being, making comparative analysis challenging. Moreover, since the 2024 data has not yet been released, making analysis difficult at this time, we will proceed with follow-up research once the data is made public. Once again, we deeply appreciate your invaluable advice. Given these constraints, we have acknowledged in the study's limitations section that only the 2019 data was utilized. We plan to conduct follow-up research comparing 2019 data with future datasets as they become available.

<Minor Issues>

1. Considering the international readership of the paper, it would be clearer to specify "Korea's survey" rather than using the general term "national statistical survey."

- Thank you for your feedback. We have replaced the term "national statistical survey" with "Korean survey" throughout the manuscript. (page 3~23).

2. There is redundancy between lines 113–127. To avoid repetition, the second paragraph in lines 120–123 should be revised.

- Thank you for your feedback. We have revised the content of the second paragraph in lines 120–123 to eliminate redundancy. (144-152 lines on page 6).

3. It seems unnecessary to specify the name of the software company. If retained, adding a hyperlink would be more appropriate.

- Thank you for your feedback. We have removed all instances of the software company names. (269 lines on page 8 / 319 lines on page 10).

4. The sentence "For this reason, ethical approval for the study" is incomplete.

- Thank you for your feedback. We have revised all incomplete sentences to ensure coherence throughout the text. (242~261 lines on page 8).

5. The explanation of NetMiner software appears multiple times, making it necessary to remove redundant descriptions.

- Thank you for your comment. We have revised the manuscript to ensure that the explanation of the software is not repeated. (line 269 on page 8 / line 319 on page 10).

6. When discussing network density, it would be more precise to use terms like "proportion" or " density" instead of "number of connections."

- Thank you for your feedback. We have revised any inaccurate content to ensure clarity and accuracy. (Table 1 on page 9).

7. Typographical errors in Table 6 need to be corrected: "cultrual" → "cultural", "activitie" → "activities".

- Thank you for your feedback. We have corrected all typographical errors accordingly. (page 13~15).

8. The paper concludes that the low-SWB group prefers personal leisure activities. However, since screen-based activities dominate in both groups, this conclusion should be reconsidered.

- Thank you for your insightful comment. We have revised and supplement the discussion section by incorporating content on screen-based activities. (lines 620–674 on page 19).

9. Instead of "adolescents with high SWB", using a more concise term like "high-SWB group"would make the sentence more succinct.

- Thank you for your feedback. We have revised the entire text to ensure conciseness. (page 3~23).

10. Vector-based images are recommended for better quality and clarity.

- Thank you for your valuable feedback. In this study, we used high-resolution raster image formats(PNG, JPEG, TIFF, etc.) to maintain the precision of data visualization while ensuring readability in most publishing and print environments. In particular, for complex network visualizations generated in Social Network Analysis(SNA), vector formats(SVG, EPS, PDF, etc.) may not always guarantee optimal readability across all platforms. Based on your suggestions, we will take them into consideration for further refinements in future research.

11. In Figure 1, if steps 4 and 5 occur simultaneously, it would be better to represent them as a parallel process rather than a sequential flow.

- Thank you for your comment. We have revised Figure 1 to combine and explain steps 4 and 5 accordingly. (line 296 on page 9).

12. Figure 2 could be integrated into Figures 3 and 4 as a legend to improve visualization.

- Thank you for your comment. We have merged Figure 2 into Figures 3 and 4 to use it as a legend. (line 494 on page 16).

13. Figures 3–6 would be easier to compare if arranged in a 2x2 layout.

- Thank you for your comment. We have rearranged Figures 3–6 into a 2x2 layout to facilitate comparison. (line 403 on page 13 / line 425 on page 14 / line 494 on page 16 / line 562 on page 18).

[Reviewer 2]

Some limitations may be known to researchers before the study begins, while others may emerge as the study progresses. Regardless of whether these constraints were anticipated or resulted from the methodology or study design, they should be clearly identified and addressed in the Discussion section, specifically in the final part of the research report. Most academic journals require authors to specify potential limitations, and many request the inclusion of a "Limitations section" at the end of the paper.

- Thank you for your review and valuable feedback on our study. We have added and organized a Limitations section at the end of the Discussion section to address this point. Once again, we deeply appreciate your insightful comments. (lines 800–851 on page 22).

[Reviewer 3]

This study analyzes the impact of leisure activities on the subjective well-being (SWB)of Korean adolescents. The authors argue that adolescents with high and low SWB exhibit different network structures. However, the network construction and analysis, which are crucial aspects of this study, are insufficiently detailed. Therefore, significant revisions are requested to include more detailed explanations.

1. Clarify the network construction method (lines 142–160).

● A more explicit explanation is needed on how the network was constructed.

● Clearly describe how links were established within the network (e.g., whether a connection was created if at least one person engaged in the same leisure activity).

● Explain how link weights were assigned and how the frequency of leisure activities influenced the network construction.

- Thank you for your insightful comments. We have supplemented the explanation in the "Research Methods – Research Procedure" section with more details on the network construction. (lines 265–277 on page 8).

2. Process of Converting a 2-Mode Network to a 1-Mode Network (lines 168–170).

● Provide a detailed explanation of the conversion process and the methodology used.

● Clearly specify the types of nodes used in the 2-mode networkand explain the criteria for removing redundant connections.

- Thank you for your insightful comments. We have added a more detailed explanation of the network conversion process to supplement and clarify this section. (lines 268–291 on page 8).

3. Please review the definition of network density in Table 1.In Table 1, density is defined as the "proportion of connected nodes,"but it is generally defined as "the ratio of actual connections to the maximum possible connections."

- Thank you for your insightful comments. We have revised the definition of density in Table 1 to accurately reflect its role as a network distribution characteristic indicator. (Table 1 on page 9).

4. Review the Network Visualizations (Figures 3–6).

● The results in Table 3 do not appear to be consistent with the network visualizations.

● For example, Table 3 mentions the presence of isolated nodes, but they are not visible in the visualizations.

● Additionally, considering the average

---

## [Decision Letter · Decision Letter 1]

31 Mar 2025

An Analysis of Adolescent Leisure Activity Structure Based on Subjective Well-being: Focusing on Social Network Analysis

PONE-D-24-52060R1

Dear Dr. Park,

We’re pleased to inform you that your manuscript has been judged scientifically suitable for publication and will be formally accepted for publication once it meets all outstanding technical requirements.

Kind regards,

Javier Fagundo-Rivera, PhD

Academic Editor

PLOS ONE

**Additional Editor Comments:**

Dear Authors,

The reviewers' comments have been fully addressed and the manuscript can be published.

Congratulations.

Reviewers' comments:

Reviewer's Responses to Questions

**Comments to the Author**

1. If the authors have adequately addressed your comments raised in a previous round of review and you feel that this manuscript is now acceptable for publication, you may indicate that here to bypass the “Comments to the Author” section, enter your conflict of interest statement in the “Confidential to Editor” section, and submit your "Accept" recommendation.

Reviewer #1: All comments have been addressed

Reviewer #2: All comments have been addressed

2. Is the manuscript technically sound, and do the data support the conclusions?

Reviewer #1: Yes

Reviewer #2: Yes

3. Has the statistical analysis been performed appropriately and rigorously? 

Reviewer #1: I Don't Know

Reviewer #2: Yes

4. Have the authors made all data underlying the findings in their manuscript fully available?

Reviewer #1: Yes

Reviewer #2: Yes

5. Is the manuscript presented in an intelligible fashion and written in standard English?

Reviewer #1: Yes

Reviewer #2: Yes

6. Review Comments to the Author

**Reviewer #1: ** I'd like to congratulate the authors for their effort to amend all the suggested changes in order to improve their manuscript.

**Reviewer #2: ** Results indicate unique leisure activity networks. High-SWB kids played more sports and socialized than low-SWB teens. Low-SWB networks were fragmented, while high-SWB networks were diversified and well-connected. Screen-based activities socialized high-SWB teens but isolated low-SWB teens. This study found SWB influences leisure engagement. Diverse and interactive leisure activities improve teenage well-being, suggesting policy and intervention changes.

7. PLOS authors have the option to publish the peer review history of their article (what does this mean? ). If published, this will include your full peer review and any attached files.

**Do you want your identity to be public for this peer review?** For information about this choice, including consent withdrawal, please see our Privacy Policy .

Reviewer #1: No

Reviewer #2: No

---

## [Editor Report · Acceptance letter]

PONE-D-24-52060R1

PLOS ONE

Dear Dr. Park,

I'm pleased to inform you that your manuscript has been deemed suitable for publication in PLOS ONE. Congratulations! Your manuscript is now being handed over to our production team.

Kind regards,

on behalf of

Dr. Javier Fagundo-Rivera

Academic Editor

PLOS ONE